# Multi-Scale Dense Networks for Resource Efficient Image Classification

**Gao Huang**
Cornell University

**Danlu Chen**
Fudan University

**Tianhong Li**
Tsinghua University

**Felix Wu**
Cornell University

**Laurens van der Maaten**
Facebook AI Research

**Kilian Weinberger**
Cornell University

## ABSTRACT

In this paper we investigate image classification with computational resource limits at test time. Two such settings are: 1. *anytime classification*, where the network's prediction for a test example is progressively updated, facilitating the output of a prediction at any time; and 2. *budgeted batch classification*, where a fixed amount of computation is available to classify a set of examples that can be spent unevenly across "easier" and "harder" inputs. In contrast to most prior work, such as the popular Viola and Jones algorithm, our approach is based on convolutional neural networks. We train multiple classifiers with varying resource demands, which we adaptively apply during test time. To maximally re-use computation between the classifiers, we incorporate them as early-exits into a single deep convolutional neural network and inter-connect them with dense connectivity. To facilitate high quality classification early on, we use a two-dimensional multi-scale network architecture that maintains coarse and fine level features all-throughout the network. Experiments on three image-classification tasks demonstrate that our framework substantially improves the existing state-of-the-art in both settings.

## 1 INTRODUCTION

Recent years have witnessed a surge in demand for applications of visual object recognition, for instance, in self-driving cars (Bojarski et al., 2016) and content-based image search (Wan et al., 2014). This demand has in part been fueled through the promise generated by the astonishing progress of convolutional networks (CNNs) on visual object recognition benchmark competition datasets, such as ILSVRC (Deng et al., 2009) and COCO (Lin et al., 2014), where state-of-the-art models may have even surpassed human-level performance (He et al., 2015; 2016).

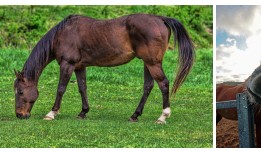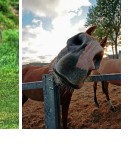

However, the requirements of such competitions differ from real-world applications, which tend to incentivize resource-hungry models with high computational demands at inference time. For example, the COCO 2016 competition was won by a large ensemble of computationally intensive CNNs[1] — a model likely far too computationally expensive for any resource-aware application. Although much smaller models would also obtain decent error, very large, computationally intensive models seem necessary to correctly classify the hard examples that make up the bulk of the remaining misclassifications of modern algorithms. To illustrate this point, Figure 1 shows two images of horses. The left image depicts a horse in canonical pose and is easy to classify, whereas the right image is taken from a rare viewpoint and is likely in the tail of the data distribution. Computationally intensive models are needed to classify such tail examples correctly, but are wasteful when applied to canonical images such as the left one.

Figure 1: Two images containing a *horse*. The left image is canonical and easy to detect even with a small model, whereas the right image requires a computationally more expensive network architecture. (Copyright Pixel Addict and Doyle (CC BY-ND 2.0).)

In real-world applications, computation directly translates into power consumption, which should be minimized for environmental and economical reasons, and is a scarce commodity on mobile

---

[1] http://image-net.org/challenges/talks/2016/GRMI-COCO-slidedeck.pdf

devices. This begs the question: *why do we choose between either wasting computational resources by applying an unnecessarily computationally expensive model to easy images, or making mistakes by using an efficient model that fails to recognize difficult images?* Ideally, our systems should automatically use small networks when test images are easy or computational resources limited, and use big networks when test images are hard or computation is abundant.

Such systems would be beneficial in at least two settings with computational constraints at test-time: *anytime prediction*, where the network can be forced to output a prediction at any given point in time; and *budgeted batch classification*, where a fixed computational budget is shared across a large set of examples which can be spent unevenly across "easy" and "hard" examples. A practical use-case of anytime prediction is in mobile apps on Android devices: in 2015, there existed $24,093$ distinct Android devices[2], each with its own distinct computational limitations. It is infeasible to train a different network that processes video frame-by-frame at a fixed framerate for each of these devices. Instead, you would like to train a single network that maximizes accuracy on all these devices, within the computational constraints of that device. The budget batch classification setting is ubiquitous in large-scale machine learning applications. Search engines, social media companies, on-line advertising agencies, all must process large volumes of data on limited hardware resources. For example, as of 2010, Google Image Search had over 10 Billion images indexed[3], which has likely grown to over 1 Trillion since. Even if a new model to process these images is only 1/10s slower per image, this additional cost would add 3170 years of CPU time. In the budget batch classification setting, companies can improve the average accuracy by reducing the amount of computation spent on "easy" cases to save up computation for "hard" cases.

Motivated by prior work in computer vision on resource-efficient recognition (Viola & Jones, 2001), we aim to develop CNNs that "slice" the computation and process these slices one-by-one, stopping the evaluation once the CPU time is depleted or the classification sufficiently certain (through "early exits"). Unfortunately, the architecture of CNNs is inherently at odds with the introduction of early exits. CNNs learn the data representation and the classifier jointly, which leads to two problems with early exits: 1. The features in the last layer are extracted directly to be used by the classifier, whereas earlier features are not. The inherent dilemma is that different kinds of features need to be extracted depending on how many layers are left until the classification. 2. The features in different layers of the network may have different scale. Typically, the first layers of a deep nets operate on a fine scale (to extract low-level features), whereas later layers transition (through pooling or strided convolution) to coarse scales that allow global context to enter the classifier. Both scales are needed but happen at different places in the network.

We propose a novel network architecture that addresses both of these problems through careful design changes, allowing for resource-efficient image classification. Our network uses a cascade of intermediate classifiers throughout the network. The first problem, of classifiers altering the internal representation, is addressed through the introduction of dense connectivity (Huang et al., 2017). By connecting all layers to all classifiers, features are no longer dominated by the most imminent early-exit and the trade-off between early or later classification can be performed elegantly as part of the loss function. The second problem, the lack of coarse-scale features in early layers, is addressed by adopting a multi-scale network structure. At each layer we produce features of all scales (fine-to-coarse), which facilitates good classification early on but also extracts low-level features that only become useful after several more layers of processing. Our network architecture is illustrated in Figure 2, and we refer to it as *Multi-Scale DenseNet (MSDNet)*.

We evaluate MSDNets on three image-classification datasets. In the anytime classification setting, we show that it is possible to provide the ability to output a prediction at any time while maintain high accuracies throughout. In the budget batch classification setting we show that MSDNets can be effectively used to adapt the amount of computation to the difficulty of the example to be classified, which allows us to reduce the computational requirements of our models drastically whilst performing on par with state-of-the-art CNNs in terms of overall classification accuracy. To our knowledge this is the first deep learning architecture of its kind that allows dynamic resource adaptation with a single model and obtains competitive results throughout.

---

[2]Source: `https://opensignal.com/reports/2015/08/android-fragmentation/`
[3]`https://en.wikipedia.org/wiki/Google_Images`

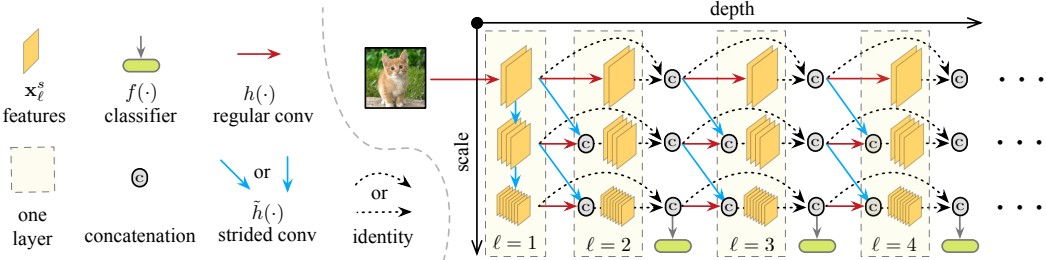

Figure 2: Illustration of the first four layers of an MSDNet with three scales. The horizontal direction corresponds to the layer direction (depth) of the network. The vertical direction corresponds to the scale of the feature maps. Horizontal arrows indicate a regular convolution operation, whereas diagonal and vertical arrows indicate a strided convolution operation. Classifiers only operate on feature maps at the coarsest scale. Connections across more than one layer are not drawn explicitly: they are implicit through recursive concatenations.

## 2    RELATED WORK

We briefly review related prior work on computation-efficient networks, memory-efficient networks, and resource-sensitive machine learning, from which our network architecture draws inspiration.

**Computation-efficient networks.** Most prior work on (convolutional) networks that are computationally efficient at test time focuses on reducing model size after training. In particular, many studies propose to prune weights (LeCun et al., 1989; Hassibi et al., 1993; Li et al., 2017) or quantize weights (Hubara et al., 2016; Rastegari et al., 2016) during or after training. These approaches are generally effective because deep networks often have a substantial number of redundant weights that can be pruned or quantized without sacrificing (and sometimes even improving) performance. Prior work also studies approaches that directly learn *compact* models with less parameter redundancy. For example, the knowledge-distillation method (Bucilua et al., 2006; Hinton et al., 2014) trains small student networks to reproduce the output of a much larger teacher network or ensemble. Our work differs from those approaches in that we train a single model that trades off computation for accuracy at test time without any re-training or finetuning. Indeed, weight pruning and knowledge distillation can be used in combination with our approach, and may lead to further improvements.

**Resource-efficient machine learning.** Various prior studies explore computationally efficient variants of traditional machine-learning models (Viola & Jones, 2001; Grubb & Bagnell, 2012; Karayev et al., 2014; Trapeznikov & Saligrama, 2013; Xu et al., 2012; 2013; Nan et al., 2015; Wang et al., 2015). Most of these studies focus on how to incorporate the computational requirements of computing particular features in the training of machine-learning models such as (gradient-boosted) decision trees. Whilst our study is certainly inspired by these results, the architecture we explore differs substantially: most prior work exploits characteristics of machine-learning models (such as decision trees) that do not apply to deep networks. Our work is possibly most closely related to recent work on FractalNets (Larsson et al., 2017), which can perform anytime prediction by progressively evaluating subnetworks of the full network. FractalNets differ from our work in that they are not explicitly optimized for computation efficiency and consequently our experiments show that MSDNets substantially outperform FractalNets. Our dynamic evaluation strategy for reducing batch computational cost is closely related to the the *adaptive computation time* approach (Graves, 2016; Figurnov et al., 2016), and the recently proposed method of adaptively evaluating neural networks (Bolukbasi et al., 2017). Different from these works, our method adopts a specially designed network with multiple classifiers, which are jointly optimized during training and can directly output confidence scores to control the evaluation process for each test example. The *adaptive computation time* method (Graves, 2016) and its extension (Figurnov et al., 2016) also perform adaptive evaluation on test examples to save batch computational cost, but focus on skipping units rather than layers. In (Odena et al., 2017), a "composer"model is trained to construct the evaluation network from a set of sub-modules for each test example. By contrast, our work uses a single CNN with multiple intermediate classifiers that is trained end-to-end. The Feedback Networks (Zamir et al., 2016) enable early predictions by making predictions in a recurrent fashion, which heavily shares parameters among classifiers, but is less efficient in sharing computation.

**Related network architectures.** Our network architecture borrows elements from neural fabrics (Saxena & Verbeek, 2016) and others (Zhou et al., 2015; Jacobsen et al., 2017; Ke et al., 2016)

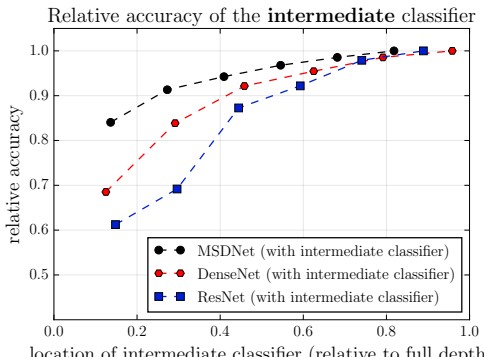 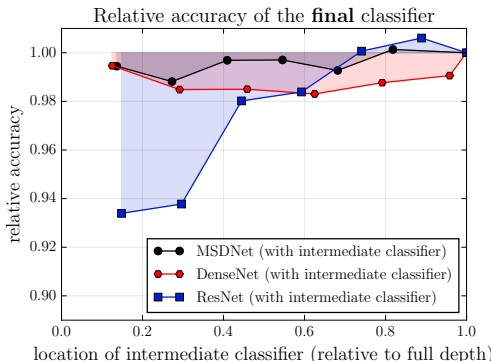

Figure 3: Relative accuracy of the intermediate classifier (*left*) and the final classifier (*right*) when introducing a single intermediate classifier at different layers in a ResNet, DenseNet and MSDNet. All experiments were performed on the CIFAR-100 dataset. Higher is better.

to rapidly construct a low-resolution feature map that is amenable to classification, whilst also maintaining feature maps of higher resolution that are essential for obtaining high classification accuracy. Our design differs from the neural fabrics (Saxena & Verbeek, 2016) substantially in that MSDNets have a reduced number of scales and no sparse channel connectivity or up-sampling paths. MSDNets are at least one order of magnitude more efficient and typically more accurate — for example, an MSDNet with less than 1 million parameters obtains a test error below 7.0% on CIFAR-10 (Krizhevsky & Hinton, 2009), whereas Saxena & Verbeek (2016) report 7.43% with over 20 million parameters. We use the same feature-concatenation approach as DenseNets (Huang et al., 2017), which allows us to bypass features optimized for early classifiers in later layers of the network. Our architecture is related to deeply supervised networks (Lee et al., 2015) in that it incorporates classifiers at multiple layers throughout the network. In contrast to all these prior architectures, our network is specifically designed to operate in resource-aware settings.

## 3 PROBLEM SETUP

We consider two settings that impose computational constraints at prediction time.

**Anytime prediction.** In the anytime prediction setting (Grubb & Bagnell, 2012), there is a finite computational budget $B > 0$ available for each test example $\mathbf{x}$. The computational budget is nondeterministic, and varies per test instance. It is determined by the occurrence of an event that requires the model to output a prediction immediately. We assume that the budget is drawn from some joint distribution $P(\mathbf{x}, B)$. In some applications $P(B)$ may be independent of $P(\mathbf{x})$ and can be estimated. For example, if the event is governed by a Poisson process, $P(B)$ is an exponential distribution. We denote the loss of a model $f(\mathbf{x})$ that has to produce a prediction for instance $\mathbf{x}$ within budget $B$ by $L(f(\mathbf{x}), B)$. The goal of an anytime learner is to minimize the expected loss under the budget distribution: $L(f) = \mathbb{E}\left[L(f(\mathbf{x}), B)\right]_{P(\mathbf{x}, B)}$. Here, $L(\cdot)$ denotes a suitable loss function. As is common in the empirical risk minimization framework, the expectation under $P(\mathbf{x}, B)$ may be estimated by an average over samples from $P(\mathbf{x}, B)$.

**Budgeted batch classification.** In the budgeted batch classification setting, the model needs to classify a set of examples $\mathcal{D}_{test} = \{\mathbf{x}_1, \ldots, \mathbf{x}_M\}$ within a finite computational budget $B > 0$ that is known in advance. The learner aims to minimize the loss across all examples in $\mathcal{D}_{test}$ within a cumulative cost bounded by $B$, which we denote by $L(f(\mathcal{D}_{test}), B)$ for some suitable loss function $L(\cdot)$. It can potentially do so by spending less than $\frac{B}{M}$ computation on classifying an "easy" example whilst using more than $\frac{B}{M}$ computation on classifying a "difficult" example. Therefore, the budget $B$ considered here is a *soft constraint* when we have a large batch of testing samples.

## 4 MULTI-SCALE DENSE CONVOLUTIONAL NETWORKS

A straightforward solution to the two problems introduced in Section 3 is to train multiple networks of increasing capacity, and sequentially evaluate them at test time (as in Bolukbasi et al. (2017)). In the anytime setting the evaluation can be stopped at any point and the most recent prediction is returned. In the batch setting, the evaluation is stopped prematurely the moment a network classifies

the test sample with sufficient confidence. When the resources are so limited that the execution is terminated after the first network, this approach is optimal because the first network is trained for exactly this computational budget without compromises. However, in both settings, this scenario is rare. In the more common scenario where some test samples can require more processing time than others the approach is far from optimal because previously learned features are never re-used across the different networks.

An alternative solution is to build a deep network with a cascade of classifiers operating on the features of internal layers: in such a network features computed for an earlier classifier can be re-used by later classifiers. However, naïvely attaching intermediate early-exit classifiers to a state-of-the-art deep network leads to poor performance.

There are two reasons why intermediate early-exit classifiers hurt the performance of deep neural networks: early classifiers lack coarse-level features and classifiers throughout interfere with the feature generation process. In this section we investigate these effects empirically (see Figure 3) and, in response to our findings, propose the MSDNet architecture illustrated in Figure 2.

**Problem: The lack of coarse-level features.** Traditional neural networks learn features of fine scale in early layers and coarse scale in later layers (through repeated convolution, pooling, and strided convolution). Coarse scale features in the final layers are important to classify the content of the whole image into a single class. Early layers lack coarse-level features and early-exit classifiers attached to these layers will likely yield unsatisfactory high error rates. To illustrate this point, we attached[4] intermediate classifiers to varying layers of a ResNet (He et al., 2016) and a DenseNet (Huang et al., 2017) on the CIFAR-100 dataset (Krizhevsky & Hinton, 2009). The blue and red dashed lines in the left plot of Figure 3 show the relative accuracies of these classifiers. All three plots gives rise to a clear trend: the accuracy of a classifier is highly correlated with its position within the network. Particularly in the case of the ResNet (blue line), one can observe a visible "staircase" pattern, with big improvements after the 2nd and 4th classifiers — located right after pooling layers.

**Solution: Multi-scale feature maps.** To address this issue, MSDNets maintain a feature representation at *multiple scales* throughout the network, and *all the classifiers only use the coarse-level features*. The feature maps at a particular layer[5] and scale are computed by concatenating the results of one or two convolutions: 1. the result of a regular convolution applied on the same-scale features from the previous layer (horizontal connections) and, if possible, 2. the result of a strided convolution applied on the finer-scale feature map from the previous layer (diagonal connections). The horizontal connections preserve and progress high-resolution information, which facilitates the construction of high-quality coarse features in later layers. The vertical connections produce coarse features throughout that are amenable to classification. The dashed black line in Figure 3 shows that MSDNets substantially increase the accuracy of early classifiers.

**Problem: Early classifiers interfere with later classifiers.** The right plot of Figure 3 shows the accuracies of the *final* classifier as a function of the location of a single *intermediate* classifier, relative to the accuracy of a network without intermediate classifiers. The results show that the introduction of an intermediate classifier harms the final ResNet classifier (blue line), reducing its accuracy by up to 7%. We postulate that this accuracy degradation in the ResNet may be caused by the intermediate classifier influencing the early features to be optimized for the short-term and not for the final layers. This improves the accuracy of the immediate classifier but collapses information required to generate high quality features in later layers. This effect becomes more pronounced when the first classifier is attached to an earlier layer.

**Solution: Dense connectivity.** By contrast, the DenseNet (red line) suffers much less from this effect. Dense connectivity (Huang et al., 2017) connects each layer with all subsequent layers and allows later layers to *bypass* features optimized for the short-term, to maintain the high accuracy of the final classifier. If an earlier layer collapses information to generate short-term features, the lost information can be recovered through the direct connection to its preceding layer. The final classifier's performance becomes (more or less) independent of the location of the intermediate

---

[4]We select six evenly spaced locations for each of the networks to introduce the intermediate classifier. Both the ResNet and DenseNet have three resolution blocks; each block offers two tentative locations for the intermediate classifier. The loss of the intermediate and final classifiers are equally weighted.

[5]Here, we use the term "layer" to refer to a column in Figure 2.

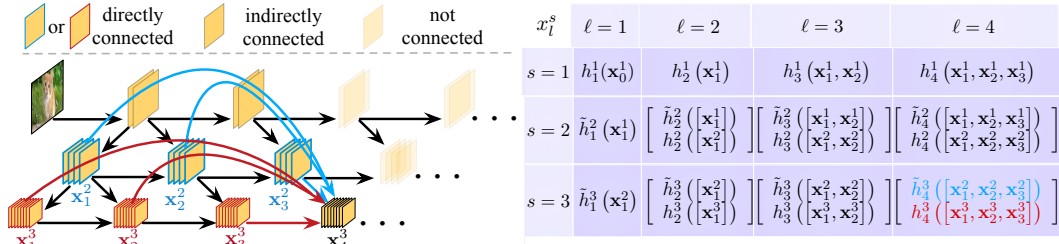

Figure 4: The output $\mathbf{x}_\ell^s$ of layer $\ell$ at the $s^{\text{th}}$ scale in a MSDNet. Herein, $[\dots]$ denotes the concatenation operator, $h_\ell^s(\cdot)$ a regular convolution transformation, and $\tilde{h}_\ell^s(\cdot)$ a strided convolutional. Note that the outputs of $h_\ell^s$ and $\tilde{h}_\ell^s$ have the same feature map size; their outputs are concatenated along the channel dimension.

classifier. As far as we know, this is the first paper that discovers that dense connectivity is an important element to early-exit classifiers in deep networks, and we make it an integral design choice in MSDNets.

### 4.1 THE MSDNET ARCHITECTURE

The MSDNet architecture is illustrated in Figure 2. We present its main components below. Additional details on the architecture are presented in Appendix A.

**First layer.** The first layer ($\ell = 1$) is unique as it includes vertical connections in Figure 2. Its main purpose is to "seed" representations on all $S$ scales. One could view its vertical layout as a miniature "S-layers" convolutional network (S=3 in Figure 2). Let us denote the output feature maps at layer $\ell$ and scale $s$ as $\mathbf{x}_\ell^s$ and the original input image as $\mathbf{x}_0^1$. Feature maps at coarser scales are obtained via down-sampling. The output $\mathbf{x}_1^s$ of the first layer is formally given in the top row of Figure 4.

**Subsequent layers.** Following Huang et al. (2017), the output feature maps $\mathbf{x}_\ell^s$ produced at subsequent layers, $\ell > 1$, and scales, $s$, are a concatenation of transformed feature maps from all previous feature maps of scale $s$ and $s-1$ (if $s > 1$). Formally, the $\ell$-th layer of our network outputs a set of features at $S$ scales $\{\mathbf{x}_\ell^1, \dots, \mathbf{x}_\ell^S\}$, given in the last row of Figure 4.

**Classifiers.** The classifiers in MSDNets also follow the dense connectivity pattern within the coarsest scale, $S$, i.e., the classifier at layer $\ell$ uses all the features $[\mathbf{x}_1^S, \dots, \mathbf{x}_\ell^S]$. Each classifier consists of two convolutional layers, followed by one average pooling layer and one linear layer. In practice, we only attach classifiers to some of the intermediate layers, and we let $f_k(\cdot)$ denote the $k^{\text{th}}$ classifier. During testing in the *anytime* setting we propagate the input through the network until the budget is exhausted and output the most recent prediction. In the *batch budget* setting at test time, an example traverses the network and exits after classifier $f_k$ if its prediction confidence (we use the maximum value of the softmax probability as a confidence measure) exceeds a pre-determined threshold $\theta_k$. Before training, we compute the computational cost, $C_k$, required to process the network up to the $k^{\text{th}}$ classifier. We denote by $0 < q \leq 1$ a fixed *exit probability* that a sample that *reaches* a classifier will obtain a classification with sufficient confidence to exit. We assume that $q$ is constant across all layers, which allows us to compute the probability that a sample exits at classifier $k$ as: $q_k = z(1-q)^{k-1}q$, where $z$ is a normalizing constant that ensures that $\sum_k p(q_k) = 1$. At test time, we need to ensure that the overall cost of classifying all samples in $\mathcal{D}_{test}$ does not exceed our budget $B$ (in expectation). This gives rise to the constraint $|\mathcal{D}_{test}| \sum_k q_k C_k \leq B$. We can solve this constraint for $q$ and determine the thresholds $\theta_k$ on a validation set in such a way that approximately $|\mathcal{D}_{test}| q_k$ validation samples exit at the $k^{\text{th}}$ classifier.

**Loss functions.** During training we use cross entropy loss functions $L(f_k)$ for all classifiers and minimize a weighted cumulative loss: $\frac{1}{|\mathcal{D}|} \sum_{(\mathbf{x},y) \in \mathcal{D}} \sum_k w_k L(f_k)$. Herein, $\mathcal{D}$ denotes the training set and $w_k \geq 0$ the weight of the $k$-th classifier. If the budget distribution $P(B)$ is known, we can use the weights $w_k$ to incorporate our prior knowledge about the budget $B$ in the learning. Empirically, we find that using the same weight for all loss functions (*i.e.*, setting $\forall k : w_k = 1$) works well in practice.

**Network reduction and lazy evaluation.** There are two straightforward ways to further reduce the computational requirements of MSDNets. First, it is inefficient to maintain all the finer scales until

the last layer of the network. One simple strategy to reduce the size of the network is by splitting it into $S$ blocks along the depth dimension, and only keeping the coarsest $(S - i + 1)$ scales in the $i^{\text{th}}$ block (a schematic layout of this structure is shown in Figure 9). This reduces computational cost for both training and testing. Every time a scale is removed from the network, we add a transition layer between the two blocks that merges the concatenated features using a $1 \times 1$ convolution and cuts the number of channels in half before feeding the fine-scale features into the coarser scale via a strided convolution (this is similar to the DenseNet-BC architecture of Huang et al. (2017)). Second, since a classifier at layer $\ell$ only uses features from the coarsest scale, the finer feature maps in layer $\ell$ (and some of the finer feature maps in the previous $S-2$ layers) do not influence the prediction of that classifier. Therefore, we group the computation in "diagonal blocks" such that we only propagate the example along paths that are required for the evaluation of the next classifier. This minimizes unnecessary computations when we need to stop because the computational budget is exhausted. We call this strategy *lazy evaluation*.

## 5 EXPERIMENTS

We evaluate the effectiveness of our approach on three image classification datasets, i.e., the CIFAR-10, CIFAR-100 (Krizhevsky & Hinton, 2009) and ILSVRC 2012 (ImageNet; Deng et al. (2009)) datasets. Code to reproduce all results is available at `https://anonymous-url`. Details on architectural configurations of MSDNets are described in Appendix A.

**Datasets.** The two CIFAR datasets contain $50,000$ training and $10,000$ test images of $32 \times 32$ pixels; we hold out $5,000$ training images as a validation set. The datasets comprise 10 and 100 classes, respectively. We follow He et al. (2016) and apply standard data-augmentation techniques to the training images: images are zero-padded with 4 pixels on each side, and then randomly cropped to produce $32 \times 32$ images. Images are flipped horizontally with probability $0.5$, and normalized by subtracting channel means and dividing by channel standard deviations. The ImageNet dataset comprises $1,000$ classes, with a total of 1.2 million training images and 50,000 validation images. We hold out 50,000 images from the training set to estimate the confidence threshold for classifiers in MSDNet. We adopt the data augmentation scheme of He et al. (2016) at training time; at test time, we classify a $224 \times 224$ center crop of images that were resized to $256 \times 256$ pixels.

**Training Details.** We train all models using the framework of Gross & Wilber (2016). On the two CIFAR datasets, all models (including all baselines) are trained using stochastic gradient descent (SGD) with mini-batch size 64. We use Nesterov momentum with a momentum weight of 0.9 without dampening, and a weight decay of $10^{-4}$. All models are trained for 300 epochs, with an initial learning rate of 0.1, which is divided by a factor 10 after 150 and 225 epochs. We apply the same optimization scheme to the ImageNet dataset, except that we increase the mini-batch size to 256, and all the models are trained for 90 epochs with learning rate drops after 30 and 60 epochs.

### 5.1 ANYTIME PREDICTION

In the anytime prediction setting, the model maintains a progressively updated distribution over classes, and it can be forced to output its most up-to-date prediction at an arbitrary time.

**Baselines.** There exist several baseline approaches for anytime prediction: FractalNets (Larsson et al., 2017), deeply supervised networks (Lee et al., 2015), and ensembles of deep networks of *varying* or *identical* sizes. FractalNets allow for multiple evaluation paths during inference time, which vary in computation time. In the anytime setting, paths are evaluated in order of increasing computation. In our result figures, we replicate the FractalNet results reported in the original paper (Larsson et al., 2017) for reference. Deeply supervised networks introduce multiple early-exit classifiers throughout a network, which are applied on the features of the particular layer they are attached to. Instead of using the original model proposed in Lee et al. (2015), we use the more competitive ResNet and DenseNet architectures (referred to as *DenseNet-BC* in Huang et al. (2017)) as the base networks in our experiments with deeply supervised networks. We refer to these as *ResNet^{MC}* and *DenseNet^{MC}*, where $MC$ stands for *multiple classifiers*. Both networks require about $1.3 \times 10^8$ FLOPs when fully evaluated; the detailed network configurations are presented in the supplementary material. In addition, we include ensembles of ResNets and DenseNets of *varying* or *identical* sizes. At test time, the networks are evaluated sequentially (in ascending order of network size) to obtain predictions for the test data. All predictions are averaged over the evaluated classifiers. On

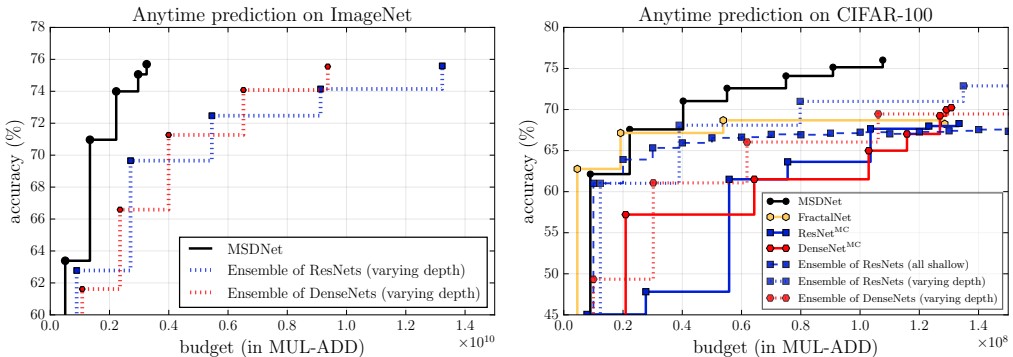

Figure 5: Accuracy (*top-1*) of *anytime prediction* models as a function of computational budget on the ImageNet (left) and CIFAR-100 (right) datasets. Higher is better.

ImageNet, we compare MSDNet against a highly competitive ensemble of ResNets and DenseNets, with depth varying from 10 layers to 50 layers, and 36 layers to 121 layers, respectively.

**Anytime prediction results** are presented in Figure 5. The left plot shows the top-1 classification accuracy on the ImageNet validation set. Here, for all budgets in our evaluation, the accuracy of MSDNet substantially outperforms the ResNets and DenseNets ensemble. In particular, when the budget ranges from $0.1 \times 10^{10}$ to $0.3 \times 10^{10}$ FLOPs, MSDNet achieves $\sim 4\% - 8\%$ higher accuracy.

We evaluate more baselines on CIFAR-100 (and CIFAR-10; see supplementary materials). We observe that MSDNet substantially outperforms ResNets[MC] and DenseNets[MC] at any computational budget within our range. This is due to the fact that after just a few layers, MSDNets have produced low-resolution feature maps that are much more suitable for classification than the high-resolution feature maps in the early layers of ResNets or DenseNets. MSDNet also outperforms the other baselines for nearly all computational budgets, although it performs on par with ensembles when the budget is very small. In the extremely low-budget regime, ensembles have an advantage because their predictions are performed by the first (small) network, which is optimized exclusively for the low budget. However, the accuracy of ensembles does not increase nearly as fast when the budget is increased. The MSDNet outperforms the ensemble as soon as the latter needs to evaluate a second model: unlike MSDNets, this forces the ensemble to repeat the computation of similar low-level features repeatedly. Ensemble accuracies saturate rapidly when all networks are shallow.

## 5.2 BUDGETED BATCH CLASSIFICATION

In budgeted batch classification setting, the predictive model receives a batch of $M$ instances and a computational budget $B$ for classifying all $M$ instances. In this setting, we use *dynamic evaluation*: we perform *early-exiting* of "easy" examples at early classifiers whilst propagating "hard" examples through the entire network, using the procedure described in Section 4.

**Baselines.** On ImageNet, we compare the dynamically evaluated MSDNet with five ResNets (He et al., 2016) and five DenseNets (Huang et al., 2017), AlexNet (Krizhevsky et al., 2012), and Google-LeNet (Szegedy et al., 2015); see the supplementary material for details. We also evaluate an ensemble of the five ResNets that uses exactly the same dynamic-evaluation procedure as MSDNets at test time: "easy" images are only propagated through the smallest ResNet-10, whereas "hard" images are classified by all five ResNet models (predictions are averaged across all evaluated networks in the ensemble). We classify batches of $M = 128$ images.

On CIFAR-100, we compare MSDNet with several highly competitive baselines, including ResNets (He et al., 2016), DenseNets (Huang et al., 2017) of varying sizes, Stochastic Depth Networks (Huang et al., 2016), Wide ResNets (Zagoruyko & Komodakis, 2016) and FractalNets (Larsson et al., 2017). We also compare MSDNet to the ResNet[MC] and DenseNet[MC] models that were used in Section 5.1, using dynamic evaluation at test time. We denote these baselines as *ResNet[MC] / DenseNet[MC] with early-exits*. To prevent the result plots from becoming too cluttered, we present CIFAR-100 results with dynamically evaluated ensembles in the supplementary material. We classify batches of $M = 256$ images at test time.

**Budgeted batch classification results** on ImageNet are shown in the left panel of Figure 7. We trained three MSDNets with different depths, each of which covers a different range of compu-

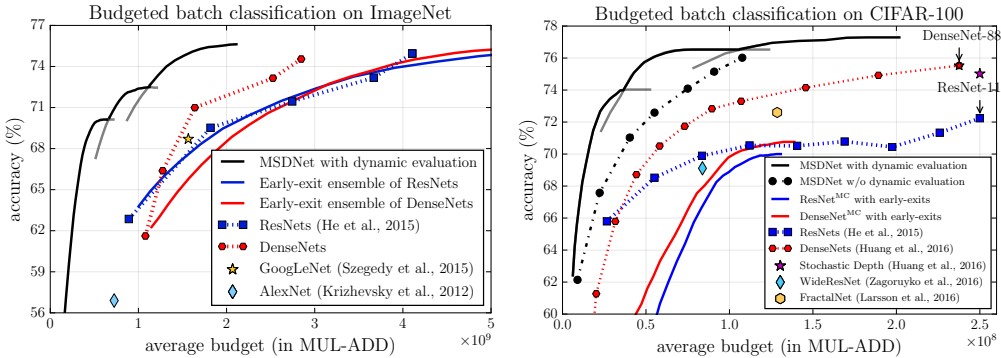

Figure 7: Accuracy (top-1) of *budgeted batch classification* models as a function of average computational budget per image the on ImageNet (left) and CIFAR-100 (right) datasets. Higher is better.

tational budgets. We plot the performance of each MSDNet as a gray curve; we select the best model for each budget based on its accuracy on the validation set, and plot the corresponding accuracy as a black curve. The plot shows that the predictions of MSDNets with dynamic evaluation are substantially more accurate than those of ResNets and DenseNets that use the same amount of computation. For instance, with an average budget of $1.7 \times 10^9$ FLOPs, MSDNet achieves a top-1 accuracy of ∼75%, which is ∼6% higher than that achieved by a ResNet with the same number of FLOPs. Compared to the computationally efficient DenseNets, MSDNet uses ∼2−3× times fewer FLOPs to achieve the same classification accuracy. Moreover, MSDNet with dynamic evaluation allows for very precise tuning of the computational budget that is consumed, which is not possible with individual ResNet or DenseNet models. The ensemble of ResNets or DenseNets with dynamic evaluation performs on par with or worse than their individual counterparts (but they do allow for setting the computational budget very precisely).

The right panel of Figure 7 shows our results on CIFAR-100. The results show that MSDNets consistently outperform all baselines across all budgets. Notably, MSDNet performs on par with a 110-layer ResNet using only 1/10th of the computational budget and it is up to ∼5 times more efficient than DenseNets, Stochastic Depth Networks, Wide ResNets, and FractalNets. Similar to results in the anytime-prediction setting, MSDNet substantially outperform ResNets$^{MC}$ and DenseNets$^{MC}$ with multiple intermediate classifiers, which provides further evidence that the coarse features in the MSDNet are important for high performance in earlier layers.

**Visualization.** To illustrate the ability of our approach to reduce the computational requirements for classifying "easy" examples, we show twelve randomly sampled test images from two ImageNet classes in Figure 6. The top row shows "easy" examples that were correctly classified and exited by the first classifier. The bottom row shows "hard" examples that would have been incorrectly classified by the first classifier but were passed on because its uncertainty was too high. The figure suggests that early classifiers recognize prototypical class examples, whereas the last classifier recognizes non-typical images.

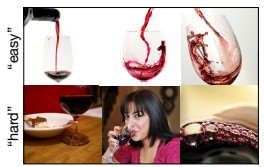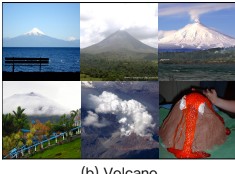

(a) Red wine      (b) Volcano

Figure 6: Sampled images from the ImageNet classes *Red wine* and *Volcano*. Top row: images exited from the first classifier of a MSDNet with correct prediction; Bottom row: images failed to be correctly classified at the first classifier but were correctly predicted and exited at the last layer.

## 5.3  MORE COMPUTATIONALLY EFFICIENT DENSENETS

Here, we discuss an interesting finding during our exploration of the MSDNet architecture. We found that following the DenseNet structure to design our network, *i.e.*, by keeping the number of output channels (or *growth rate*) the same at all scales, did not lead to optimal results in terms of the accuracy-speed trade-off. The main reason for this is that compared to network architectures like ResNets, the DenseNet structure tends to apply more filters on the high-resolution feature maps in the network. This helps to reduce the number of parameters in the model, but at the same time, it greatly increases the computational cost. We tried to modify DenseNets by doubling the growth rate

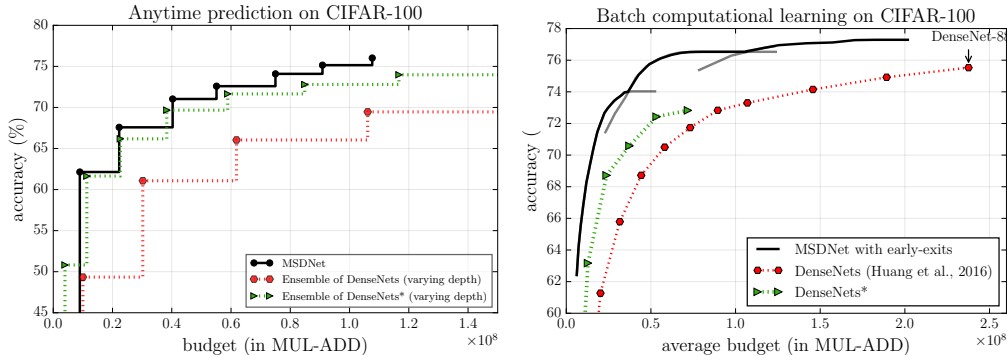

Figure 8: Test accuracy of DenseNet* on CIFAR-100 under the anytime learning setting (*left*) and the budgeted batch setting (*right*).

after each transition layer, so that more filters are applied to low-resolution feature maps. It turns out that the resulting network, which we denote as DenseNet*, significantly outperform the original DenseNet in terms of computational efficiency.

We experimented with DenseNet* in our two settings with test time budget constraints. The left panel of Figure 8 shows the anytime prediction performance of an ensemble of DenseNets* of varying depths. It outperforms the ensemble of original DenseNets of varying depth by a large margin, but is still slightly worse than MSDNets. In the budgeted batch budget setting, DenseNet* also leads to significantly higher accuracy over its counterpart under all budgets, but is still substantially outperformed by MSDNets.

## 6 CONCLUSION

We presented the MSDNet, a novel convolutional network architecture, optimized to incorporate CPU budgets at test-time. Our design is based on two high-level design principles, to generate and maintain coarse level features throughout the network and to inter-connect the layers with dense connectivity. The former allows us to introduce intermediate classifiers even at early layers and the latter ensures that these classifiers do not interfere with each other. The final design is a two dimensional array of horizontal and vertical layers, which decouples depth and feature coarseness. Whereas in traditional convolutional networks features only become coarser with increasing depth, the MSDNet generates features of all resolutions from the first layer on and maintains them throughout. The result is an architecture with an unprecedented range of efficiency. A single network can outperform all competitive baselines on an impressive range of computational budgets ranging from highly limited CPU constraints to almost unconstrained settings.

As future work we plan to investigate the use of resource-aware deep architectures beyond object classification, *e.g.* image segmentation (Long et al., 2015). Further, we intend to explore approaches that combine MSDNets with model compression (Chen et al., 2015; Han et al., 2015), spatially adaptive computation (Figurnov et al., 2016) and more efficient convolution operations (Chollet, 2016; Howard et al., 2017) to further improve computational efficiency.

### ACKNOWLEDGMENTS

The authors are supported in part by grants from the National Science Foundation ( III-1525919, IIS-1550179, IIS-1618134, S&AS 1724282, and CCF-1740822), the Office of Naval Research DOD (N00014-17-1-2175), and the Bill and Melinda Gates Foundation.

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

## A   Details of MSDNet Architecture and Baseline Networks

We use MSDNet with three scales on the CIFAR datasets, and the network reduction method introduced in 4.1 is applied. Figure 9 gives an illustration of the reduced network. The convolutional layer functions in the first layer, $h_1^s$, denote a sequence of 3×3 convolutions (Conv), batch normalization (BN; Ioffe & Szegedy (2015)), and rectified linear unit (ReLU) activation. In the computation of $\tilde{h}_1^s$, down-sampling is performed by applying convolutions using strides that are powers of two. For subsequent feature layers, the transformations $h_\ell^s$ and $\tilde{h}_\ell^s$ are defined following the design in DenseNets (Huang et al., 2017): Conv($1 \times 1$)-BN-ReLU-Conv($3 \times 3$)-BN-ReLU. We set the number of output channels of the three scales to 6, 12, and 24, respectively. Each classifier has two down-sampling convolutional layers with 128 dimensional $3\times3$ filters, followed by a $2\times2$ average pooling layer and a linear layer.

The MSDNet used for ImageNet has four scales, respectively producing 16, 32, 64, and 64 feature maps at each layer. The network reduction is also applied to reduce computational cost. The original images are first transformed by a 7×7 convolution and a 3×3 max pooling (both with stride 2), before entering the first layer of MSDNets. The classifiers have the same structure as those used for the CIFAR datasets, except that the number of output channels of each convolutional layer is set to be equal to the number of its input channels.

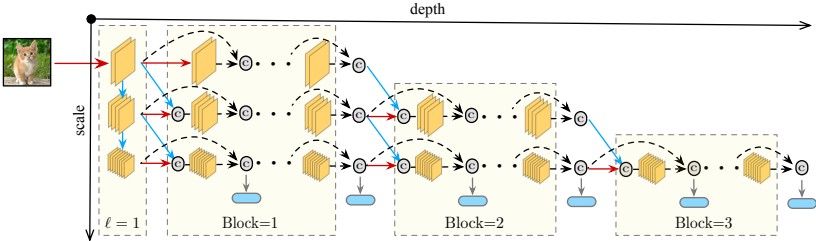

Figure 9: Illustration of an MSDNet with network reduction. The network has $S = 3$ scales, and it is divided into three blocks, which maintain a decreasing number of scales. A transition layer is placed between two contiguous blocks.

**Network architecture for anytime prediction.** The MSDNet used in our anytime-prediction experiments has 24 layers (each layer corresponds to a column in Fig. 1 of the main paper), using the reduced network with transition layers as described in Section 4. The classifiers operate on the output of the $2\times(i+1)^{\text{th}}$ layers, with $i=1,\ldots,11$. On ImageNet, we use MSDNets with four scales, and the $i^{\text{th}}$ classifier operates on the $(k\times i+3)^{\text{th}}$ layer (with $i=1,\ldots,5$ ), where $k=4,6$ and 7. For simplicity, the losses of all the classifiers are weighted equally during training.

**Network architecture for budgeted batch setting.** The MSDNets used here for the two CIFAR datasets have depths ranging from 10 to 36 layers, using the reduced network with transition layers as described in Section 4. The $k^{\text{th}}$ classifier is attached to the $(\sum_{i=1}^{k} i)^{\text{th}}$ layer. The MSDNets used for ImageNet are the same as those described for the anytime learning setting.

**ResNet$^{\text{MC}}$ and DenseNet$^{\text{MC}}$.** The *ResNet$^{MC}$* has 62 layers, with 10 residual blocks at each spatial resolution (for three resolutions): we train early-exit classifiers on the output of the $4^{\text{th}}$ and $8^{\text{th}}$ residual blocks at each resolution, producing a total of 6 intermediate classifiers (plus the final classification layer). The *DenseNet$^{MC}$* consists of 52 layers with three dense blocks and each of them has 16 layers. The six intermediate classifiers are attached to the $6^{\text{th}}$ and $12^{\text{th}}$ layer in each block, also with dense connections to all previous layers in that block.

## B   Additional Results

### B.1   Ablation Study

We perform additional experiments to shed light on the contributions of the three main components of MSDNet, *viz.*, multi-scale feature maps, dense connectivity, and intermediate classifiers.

We start from an MSDNet with six intermediate classifiers and remove the three main components one at a time. To make our comparisons fair, we keep the computational costs of the full networks similar, at around $3.0 \times 10^8$ FLOPs, by adapting the network width, i.e., number of output channels at each layer. After removing all the three components in an MSDNet, we obtain a regular VGG-like convolutional network. We show the classification accuracy of all classifiers in a model in the left panel of Figure 10. Several observations can be made: 1. the dense connectivity is crucial for the performance of MSDNet and removing it hurts the overall accuracy drastically (orange vs. black curve); 2. removing multi-scale convolution hurts the accuracy only in the lower budget regions, which is consistent with our mo-

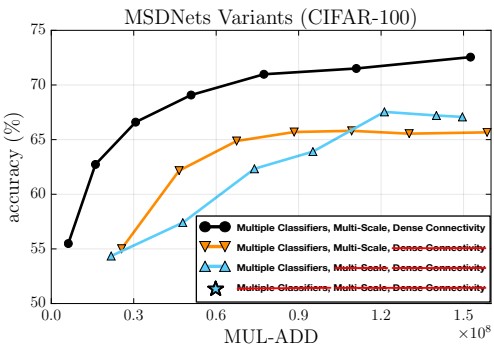

Figure 10: Ablation study (on CIFAR-100) of MS-DNets that shows the effect of dense connectivity, multi-scale features, and intermediate classifiers. Higher is better.

tivation that the multi-scale design introduces discriminative features early on; 3. the final canonical CNN (star) performs similarly as MSDNet under the specific budget that matches its evaluation cost exactly, but it is unsuited for varying budget constraints. The final CNN performs substantially better at its particular budget region than the model without dense connectivity (orange curve). This suggests that dense connectivity is particularly important in combination with multiple classifiers.

## B.2 RESULTS ON CIFAR-10

For the CIFAR-10 dataset, we use the same MSDNets and baseline models as we used for CIFAR-100, except that the networks used here have a 10-way fully connected layer at the end. The results under the anytime learning setting and the batch computational budget setting are shown in the left and right panel of Figure 11, respectively. Similar to what we have observed from the results on CIFAR-100 and ImageNet, MSDNets outperform all the baselines by a significant margin in both settings. As in the experiments presented in the main paper, ResNet and DenseNet models with multiple intermediate classifiers perform relatively poorly.

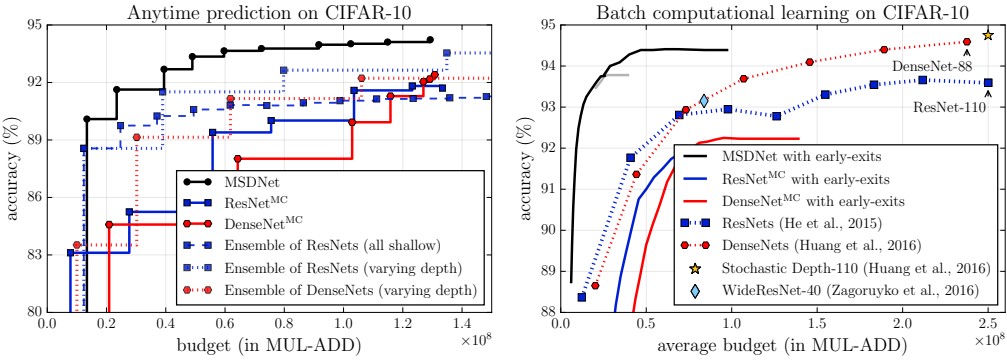

Figure 11: Classification accuracies on the CIFAR-10 dataset in the anytime-prediction setting (*left*) and the budgeted batch setting (*right*).

