# OpenReview forum: "Multi-Scale Dense Networks for Resource Efficient Image Classification"
_ICLR.cc/2018/Conference — Accept (Oral)_

### Official Review · AnonReviewer2 · 2017-11-23
**This is a well written paper that incorporates CPU budgets at test time via a multi-scale design of the DenseNet architecture.**

**Rating:** 8
**Confidence:** 4

**Review:**

This work proposes a variation of the DenseNet architecture that can cope with computational resource limits at test time. The paper is very well written, experiments are clearly presented and convincing and, most importantly, the research question is exciting (and often overlooked).

My only major concern is the degree of technical novelty with respect to the original DenseNet paper of Huang et al. (2017). The authors add a hierarchical, multi-scale structure and show that DenseNet can better cope with it than ResNet (e.g., Fig. 3). They investigate pros and cons in detail adding more valuable analysis in the appendix. However, this work is basically an extension of the DenseNet approach with a new problem statement and additional, in-depth analysis.

Some more minor comments:

-	Please enlarge Fig. 4.
-	I did not fully grasp the details in the first "Solution" paragraph on P5. Please extend and describe in more detail.

In conclusion, this is a very well written paper that designs the network architecture (of DenseNet) such that it is optimized to include CPU budgets at test time. I recommend acceptance to ICLR18.

---

> ### Author Response · Authors · 2018-01-04
> **response**
>
> Thanks for positive comments.
>
> # difference to DenseNet
> Although dense connectivity is one of the two key components in our MSDNet, this paper is quite different from the original DenseNet paper: (1) in this paper we tackle a very different problem, the inference of deep models with computational resource limits at test time; (2) we show the multi-scale features are crucial for learning accurate early classifiers. Finally, MSDNet yields 2x to 5x faster inference speed than DenseNet under the batch budgeted setting.
>
> # minors
> Thanks for these suggestions. We have incorporated them in the updated version.

---

### Official Review · AnonReviewer1 · 2017-11-27
**clear and effective method connecting image scale and evaluation times**

**Rating:** 7
**Confidence:** 4

**Review:**

This paper presents a method for image classification given test-time computational budgeting constraints.  Two problems are considered:  "any-time" classification, in which there is a time constraint to evaluate a single example, and batched budgets, in which there is a fixed budget available to classify a large batch of images.  A convolutional neural network structure with a diagonal propagation layout over depth and scale is used, so that each activation map is constructed using dense connections from both same and finer scale features.  In this way, coarse-scale maps are constructed quickly, then continuously updated with feed-forward propagation from lower layers and finer scales, so they can be used for image classification at any intermediate stage.  Evaluations are performed on ImageNet and CIFAR-100.

I would have liked to see the MC baselines also evaluated on ImageNet --- I'm not sure why they aren't there as well?  Also on p.6 I'm not entirely clear on how the "network reduction" is performed --- it looks like finer scales are progressively dropped in successive blocks, but I don't think they exactly correspond to those that would be needed to evaluate the full model (this is "lazy evaluation").  A picture would help here, showing where the depth-layers are divided between blocks.

I was also initially a bit unclear on how the procedure described for batched budgeted evaluation achieves the desired result:  It seems this relies on having a batch that is both large and varied, so that its evaluation time will converge towards the expectation.  So this isn't really a hard constraint (just an expected result for batches that are large and varied enough).  This is fine, but could perhaps be pointed out if that is indeed the case.

Overall, this seems like a natural and effective approach, and achieves good results.

---

> ### Author Response · Authors · 2018-01-04
> **Response**
>
> Thanks for the positive comments.
>
> # MC baselines on ImageNet
> We exclude these results in our current version as we observed that they are far from competitive on both CIFAR-10 and CIFAR-100. We are testing the MC baselines on ImageNet, and will include it in a later version, but won’t expect them to be strong baselines.
>
> # network reduction
> The ‘network reduction’ is a design choice to reduce redundancy in the network, while ‘lazy evaluation’ is a strategy to avoid redundant computations. We have added a figure (Figure 9) in the appendix to illustrate the reduced network as suggested.
>
> # batched budgeted evaluation
> Thanks for pointing out. We have emphasize that the notion of budget in this context is a “soft constraint” given a large batch of testing samples.

---

### Official Review · AnonReviewer3 · 2017-11-28
**Great speed-up and performance for CNN classification**

**Rating:** 10
**Confidence:** 4

**Review:**

This paper introduces a new model to perform image classification with limited computational resources at test time. The model is based on a multi-scale convolutional neural network similar to the neural fabric (Saxena and Verbeek 2016), but with dense connections (Huang et al., 2017) and with a classifier at each layer. The multiple classifiers allow for a finer selection of the amount of computation needed for a given input image. The multi-scale representation allows for better performance at early stages of the network. Finally the dense connectivity allows to reduce the negative effect that early classifiers have on the feature representation for the following layers.
A thorough evaluation on ImageNet and Cifar100 shows that the network can perform better than previous models and ensembles of previous models with a reduced amount of computation.

Pros:
- The presentation is clear and easy to follow.
- The structure of the network is clearly justified in section 4.
- The use of dense connectivity to avoid the loss of performance of using early-exit classifier is very interesting.
- The evaluation in terms of anytime prediction and budgeted batch classification can represent real case scenarios.
- Results are very promising, with 5x speed-ups and same or better accuracy that previous models.
- The extensive experimentation shows that the proposed network is better than previous approaches under different regimes.

Cons:
- Results about the more efficient densenet* could be shown in the main paper

Additional Comments:
- Why in training you used logistic loss instead of the more common cross-entropy loss? Has this any connection with the final performance of the network?
- In fig. 5 left for completeness I would like to see also results for DenseNet^MT and ResNet^MT
- In fig. 5 left I cannot find the 4% and 8% higher accuracy with 0.5x10^10 to 1.0x10^10 FLOPs, as mentioned in section 5.1 anytime prediction results
- How the budget in terms of Mul-Adds is actually estimated?

I think that this paper present a very powerful approach to speed-up the computational cost of a CNN at test time and clearly explains some of the common trade-offs between speed and accuracy and how to improve them. The experimental evaluation is complete and accurate.

---

> ### Author Response · Authors · 2018-01-04
> **Response**
>
> Thank you for the encouraging comments!
>
> # DenseNet*
> We have included the DenseNet* results in the main paper as suggested. We placed this network originally in the appendix to keep the focus of the main manuscript on the MSDNet architecture, and it was introduced for the first time in this paper (although as a competitive baseline).
>
> # logistic loss
> We actually used the cross entropy loss in our experiments. We have fixed this sentence. Thanks for pointing out.
>
> # DenseNet^MC and ResNet^MC on ImageNet (left panel of Fig.5)
> We observed that DenseNet^MC and ResNet^MC are two of the weakest baselines on both CIFAR-10 and CIFAR-100 datasets. Therefore, we thought their results on ImageNet probably won’t add much to the paper. We can add these results in a later version.
>
> # improvements in the anytime setting
> It should be 4% and 8% higher accuracy when the budget ranges from 0.1x10^10* to 0.3x10^10* FLOPs. We have corrected it in the updated version.
>
> # actually budget
> For many devices, e.g., ARM processor, the actual inference time is basically a linear function of the number of Mul-Add operations. Thus in practice, given a specific device, we can estimate the budget in terms of Mul-Add according to the real time budget.

---

### Public Comment · (anonymous) · 2018-02-07
**Reproducibility challenge report**

Summary
* We've run the actual implementation. The net worked really well, so we focused on implementing it in PyTorch and testing on other datasets (CIFAR-10 and Caltech 101).
* We've checked if training large network (with multiple classifiers) and using just a part of it during testing is possible (we can use earlier classifiers as an early-exits) and what are the results. It's actually working and yielding great results.

Link to the full report here: https://github.com/janchorowski/nn_assignments/blob/nn17_fall/project_reports/MSDNet/REPORT.md

---

### Public Comment · (anonymous) · 2018-02-27
**Reproducibility challenge report**

As our deep learning seminar project, we participated in ICLR reproducibility challenge. The project included re-implementing MSDNet paper, reproducibility of the results, and testing a new variant on the original MSDNet architecture.
Summary of our results:
1. We managed to reproduce the results presented in the paper, both with the provided implementation in Lua, and with our implementation in pytorch.
2. We tested some variants of plugging Global Convolutional Network (with separable kernel) to the features network, in order to reduce its parameters and runtime.

Link to the full report and code: https://github.com/avirambh/MSDNet-GCN/blob/master/report/MSDNet_reproducibility_report.pdf

---

### Decision · Program_Chairs · 2018-01-29
**ICLR 2018 Conference Acceptance Decision**

**Decision:**

Accept (Oral)

**Comment:**

As stated by reviewer 3 "This paper introduces a new model to perform image classification with limited computational resources at test time. The model is based on a multi-scale convolutional neural network similar to the neural fabric (Saxena and Verbeek 2016), but with dense connections (Huang et al., 2017) and with a classifier at each layer."
As stated by reviewer 2 "My only major concern is the degree of technical novelty with respect to the original DenseNet paper of Huang et al. (2017). ".  The authors assert novelty in the sense that they provide a solution to improve computational efficiency and focus on this aspect of the problem. Overall, the technical innovation is not huge, but I think this could be a very useful idea in practice.